


# A MULTI-INSTRUMENTAL AND MODELLING ANALYSIS OF THE IONOSPHERIC RESPONSES TO THE SOLAR ECLIPSE OF DECEMBER 14, 2020, OVER THE BRAZILIAN REGION

Laysa C. A. Resende[1,2*], Yajun Zhu[1], Clezio M. Denardini[2], Sony S. Chen[2], Ronan A. J. Chagas[2], Lígia A. Da Silva[1,2], Carolina S. Carmo[2], Juliano Moro[1,3], Diego Barros[2], Paulo A. B. Nogueira[4], José P. Marchezi[1,2], Giorgio A. S. Picanço[2], Paulo Jauer[1,2], Régia P. Silva[2], Douglas Silva[1,2], José A. Carrasco[2], Chi Wang[1], Zhengkuan Liu[1]

[1]State Key Laboratory of Space Weather, Beijing, China.
[2]National Institute for Space Research – INPE, São José dos Campos-SP, Brazil.
[3]Southern Space Coordination - COESU, Santa Maria-RS, Brazil.
[4]Instituto Federal de Educação Ciência e Tecnologia de São Paulo - IFSP, Jacareí, Brazil.

*Correspondence to*: Laysa C. A. Resende (laysa.resende@inpe.br; laysa.resende@gmail.com)

**Abstract.**

This work presents an analysis of the ionospheric responses to the solar eclipse that occurred on December 14, 2020, over the Brazilian sector. This event partially covers the south of Brazil, providing an excellent opportunity to study the modifications in the peculiarities that occur in this sector, as the Equatorial Ionization Anomaly (EIA). Therefore, we used the Digisonde data available in this period for two sites, Campo Grande (CG, 20.47° S, 54.60° W, dip ~23° S) and Cachoeira Paulista (CXP, 22.70° S, 45.01° W, dip ~35° S), assessing the E, and F regions, and Es layer behaviors. Additionally, a numerical model (MIRE, Portuguese acronym for E Region Ionospheric Model) is used to analyze the E layer dynamics modification around these times. The results show the F1 region disappearance and an apparent electronic density reduction in the E region during the solar eclipse. We also analyzed the total electron content (TEC) maps from the Global Navigation Satellite System (GNSS) that indicate a weakness in the EIA. On the other hand, we observe the rise of the Es layer electron density, which is related to the gravity waves strengthened during solar eclipse events. Finally, our results lead to a better understanding of the restructuring mechanisms in the ionosphere at low latitudes during the solar eclipse events, even though they only partially reached the studied regions.



# 1 Introduction

Events as a solar eclipse, where the moon passes between the Sun and the Earth, can cause modifications in the ionosphere. The solar radiation is attenuated, and, consequently, the UV solar flux decreases, affecting all the ionospheric layers (Fargues et al., 2001; Chandra et al., 2007; Vogrincic et al., 2020). Thus, it is possible to observe influences in the Total Electron Content (TEC) (Cherniak and Zakharenkova, 2018), in the Equatorial Ionospheric Anomaly (EIA) (Chen et al., 2019; Jonah et al., 2020), a decrease in the E and F region densities (Chandra et al., 2007), and changes in all types of Sporadic E (Es) layers (Adeniyi et al., 2007; Pezzopane et al., 2015).

Many studies about the ionosphere response in partial or total solar eclipse were performed in the last years. Sridharan et al. (2002) analyzed the ionosphere electrodynamics during the solar eclipse on August 11, 1999, over the equatorial station in Trivandrum (8.5° N, 77° E, dip 0.5° N), India. Their results showed some characteristics in the ionograms as intense blanketing Es layer ($Es_b$) occurrence, and an increase in the F region virtual height ($h'F$) after the solar eclipse, emerging the spread-F structures. The authors concluded that the solar eclipse could lead to favorable conditions for irregularity development. Chernogor et al. (2019) recently analyzed the solar eclipse effects in the mid-latitude daytime ionospheric plasma. The eclipse event occurred along with the magnetic storm recovery phase on March 20, 2015. However, the authors concluded that the increases in the F region peak height ($hmF2$) during the maximum solar occultation and the decrease in the electron density around 190-210 km are consequences of the solar eclipse.

Cherniak and Zakharenkova (2018) and Chen et al. (2019a) studied the total eclipse of August 21, 2017, in American sector. Both studies analyzed the TEC maps during the solar eclipse. They found a TEC decrease by ~30-40% along the totality path within an area of 75% obscuration. In fact, their work showed that the vertical electron density latitudinal variations presented enhancements or reductions of EIA crests depending on the latitude.

Regarding the Es layer behavior, Chen et al. (2010a), Chen et al. (2010b), and Tiwari et al. (2019) showed a considerable enhancement in their electronic density, meaning that an intensification of the Es





layer occurred during the total solar eclipse on July 22, 2009. Pezzopane et al. (2015) analyzed the Es layer using the ionosondes located at mid-latitudes stations of Italy during the solar eclipse that occurred on March 20, 2015. They found that the solar eclipse affects the temporal persistence of the Es layer. In all these works, the Es layer changes were attributed to gravity wave occurrences caused by thermal

gradients related to the solar eclipse event. On the other hand, Chen et al. (2019b) investigated the Es layer response during the solar eclipse in the American Continent on August 21, 2017. They found an intensity reduction of these layers during this event, which they associated with the photoionization decrease.

        Martínez-Ledesma et al. (2020) predicted the F region behavior during the total solar eclipse on

December 14, 2020. They used the Sheffield University Plasmasphere Ionosphere Model (SUPIM-INPE) (Bailey et al., 1993; Souza et al., 2010) to evaluate the TEC modifications at low latitudes. The predictions expected a TEC decrease of up to 22% in regions along the path of totality. Also, the simulations showed a minor TEC reduction around the magnetic equator locations that even so affected the fountain effect, and consequently, the EIA crests.

In this work, we perform a multi-instrumental and modeling analysis of the ionospheric response over low latitudes in the Brazilian regions predicted by Martínez-Ledesma et al. (2020) for the solar eclipse event on December 14, 2020. We used the Digisonde data to observe the modifications in the E and F regions, and the Es layers over two sites, Campo Grande (CG, 20.47° S, 54.6° W, dip ~23° S) and Cachoeira Paulista (CXP, 22.70° S, 45.01° W, dip ~35° S). Also, a numerical model (MIRE, Portuguese

acronym for E Region Ionospheric Model) is used to analyze the E layer chemistry. The TEC maps derived from the Global Navigation Satellite System (GNSS) are used to observe changes in the EIA. Finally, the results showed that solar eclipses can cause significant ionosphere modifications even though they only partially reach the Brazilian low latitude regions.

## 2 Methodology

In the following, we briefly describe each set of data used in this study: Digisonde data, GNSS TEC variation, and MIRE model.



## 2.1 Digisonde Data

In this work, we used ionospheric parameters of the vertical electron density profiles obtained from Digisonde, called ionograms. This equipment is an HF radar that transmits radio waves continuously into the ionosphere ranging from 1 to 30 MHz (Reinisch et al., 2009). We used the Digisonde data from CXP and CG over the Brazilian sector provided by the Brazilian Studies and Monitoring of Space Weather (EMBRACE) (available online in http://www2.inpe.br/climaespacial/portal/en/).

We evaluated the F region behavior using height parameters, such as the virtual height ($h'$F) and peak height ($hm$F2), which is important to investigate the changes in this region. Also, we analyzed the frequency parameters of the F and E regions and Es layer ($fo$F2, $fo$E, and $fb$Es, respectively), which are related to the electronic density at the layer peak. The $fb$Es is the frequency at which reflection from a layer at superior heights starts to be visible in ionograms. The time resolution is 10 min from the ionograms in stations considered here.

## 2.2 TEC Analysis

The GNSS receiver data were used to obtain the total number of electrons (TEC) in a given ionospheric path. TEC is a measurement of the electrons in a column of unitary cross-section area between the satellite and the receiver. Due to the high number of stations over the Brazilian sector, it is possible to construct the two-dimensional maps of the absolute vertical TEC values ranging from 50 to ~500 km of spatial resolution in latitude and longitude, every 10 minutes (Otsuka et al., 2002; Takahashi et al., 2016). This current work analyzes the EIA during the eclipse occurrence using these maps, available online on the EMBRACE website.

## 2.3 MIRE Model

We have used a theoretical model, called MIRE, which provides the E region and Es layer electron densities as follows (Carrasco et al., 2007; Resende et al. 2017a, 2017b, 2020, 2021):

$$ne = [O_2^+] + [NO^+] + [O^+] + [N_2^+] + [Fe^+] + [Mg^+]. \tag{1}$$



This model solves a set of partial differential equations of the continuity and momentum between 00 UT and 24 UT in the height range from 86 to 120 km for the main molecular/atomic ions in the E region ($NO^+$, $O_2^+$, $N_2^+$, $O^+$), and metallic ions ($Fe^+$, $Mg^+$).

MIRE continuity equation of each constituent $N_i$:

$$\frac{\partial [N_i]}{\partial t} = P - L - \frac{\partial V_{iz} \, [N_i]}{\partial z},$$

(2)

5 is used to calculate the ion density taking into account the production ($P$), loss ($L$), and transport ($\frac{\partial V_{iz} \, [N_i]}{\partial z}$.). The transport term depends on the wind and electric field parameters (Resende et al. 2020). This analysis only considers the E-region chemistry, thus neglecting the transport terms and metallic ions. More details about the MIRE model can be found in Carrasco et al. (2007) and Resende et al. (2017a).

## 2.4 Solar Eclipse Characteristics

10 We divided this study into two ionospheric responses of the solar eclipse that occurred on December 14, 2020: (1) the ionospheric changes in the F region, and consequently EIA behavior; and (2) the E region and Es layer behavior during the solar eclipse hours. Figure 1 shows the solar eclipse evolution between 1545 UT and 1800 UT for every 15 minutes at 250 km height. The colors mean the obscuration varying from 10% (purple) to 100% (yellow). Notice that only two Digisonde stations (CXP and CG) had a solar eclipse obscuration over the Brazilian sector with available data. In CXP, the solar eclipse influence starts about 1615 UT until 1800 UT, while, in CG, it was between around 1600 UT and 1715 UT. Therefore, we analyzed these regions since the solar eclipse provides a great opportunity to study the responses of the ionospheric regions to the rapid solar radiation variation.





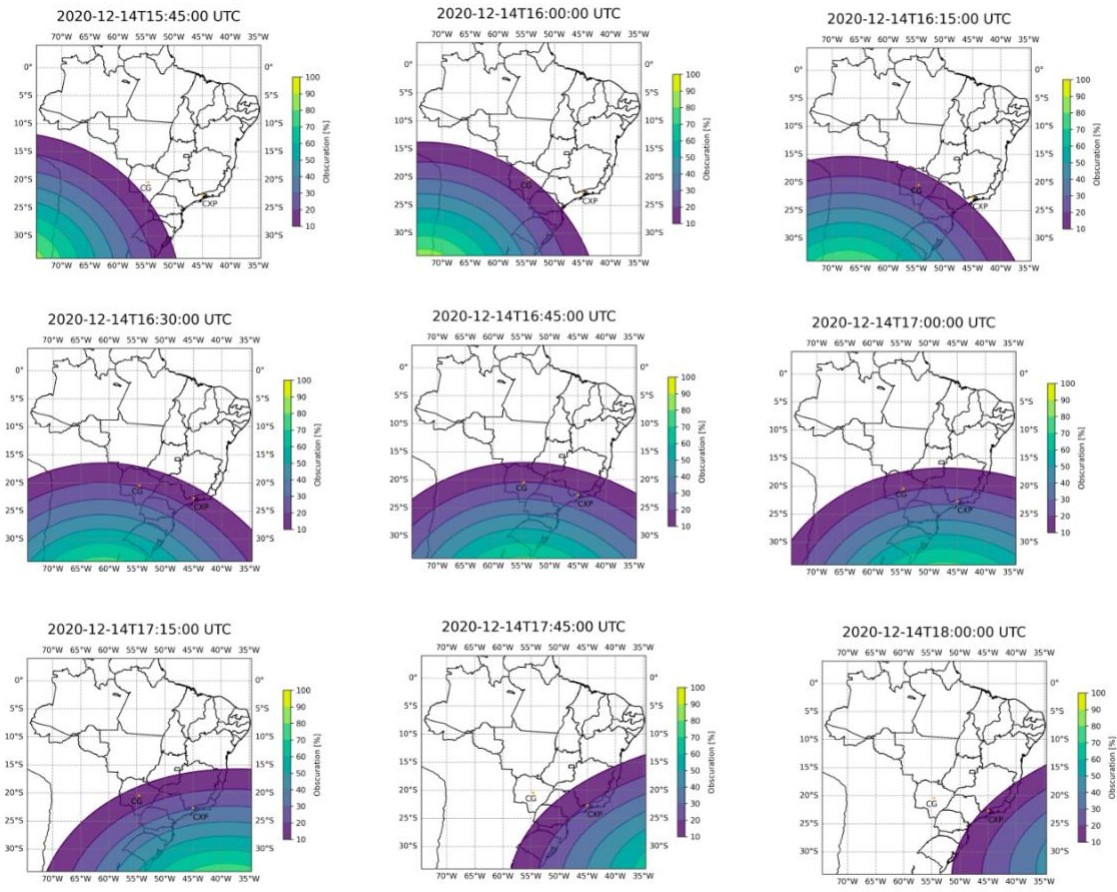

**Figure 1:** Eclipse obscuration mask at 250 km height between 1545 UT and 1800 UT for every 15 minutes on December 14, 2020. The contour colors are obscuration varying from 10% (purple) to 100% (yellow). CG and CXP are marked in these maps, referring to the Digisonde stations.

## 3 Results and Discussions

### 3.1 Space weather conditions during the eclipse 2020 versus quiet period

The interplanetary medium parameters are measured from the Proton and Alpha Monitor (SWEPAM) and Magnetic Field Experiment (MAG) instruments onboard the Advanced Composition Explorer (ACE) spacecraft (Stone et al., 1998). Figure 2 shows the solar wind speed, $V_p$ (a), proton density, $N_p$ (b), and Bz component of the Interplanetary Magnetic Field (IMF) (c) measured at the L1





Lagrangian point. Also, we show the Dst index in panel d. The data for a solar eclipse that occurred on December 14, 2020, is shown in the orange line, and we used a reference period on December 4, 2020, in the blue line.

The Vp observed in the eclipse day and the quiet period is concentrated below 400 km/s, which are considerably slow wind (Tsurutani et al., 2011; Isaacs et al., 2015). The proton density fluctuates around 2 particles/cm$^3$ during almost the entire eclipse day, except in a short period, between 1648 UT and 1912 UT, reaching ~ 5 particles/cm$^3$. This short period can be associated with the solar sector boundary-crossing (Figure not shown here). However, these values are still considered low. Also, the B$_z$ component fluctuates around zero during almost the entire eclipse day, reaching the maximum negative value of ~ -2.5 nT during a short time and presenting few positive incursions (maximum of ~ 2 nT). To confirm that the event occurred on a geomagnetically quiet day, we show the Dst index in Figure 2d. Although this parameter showed a minor enhancement during the solar eclipse event starting at 1400 UT, the values remained very low, oscillating around zero. Finally, all the interplanetary parameters showed that this day is not geomagnetically disturbed in terms of the ionosphere influence. Therefore, this event is an excellent opportunity to analyze the ionospheric influences during a solar eclipse occurrence.



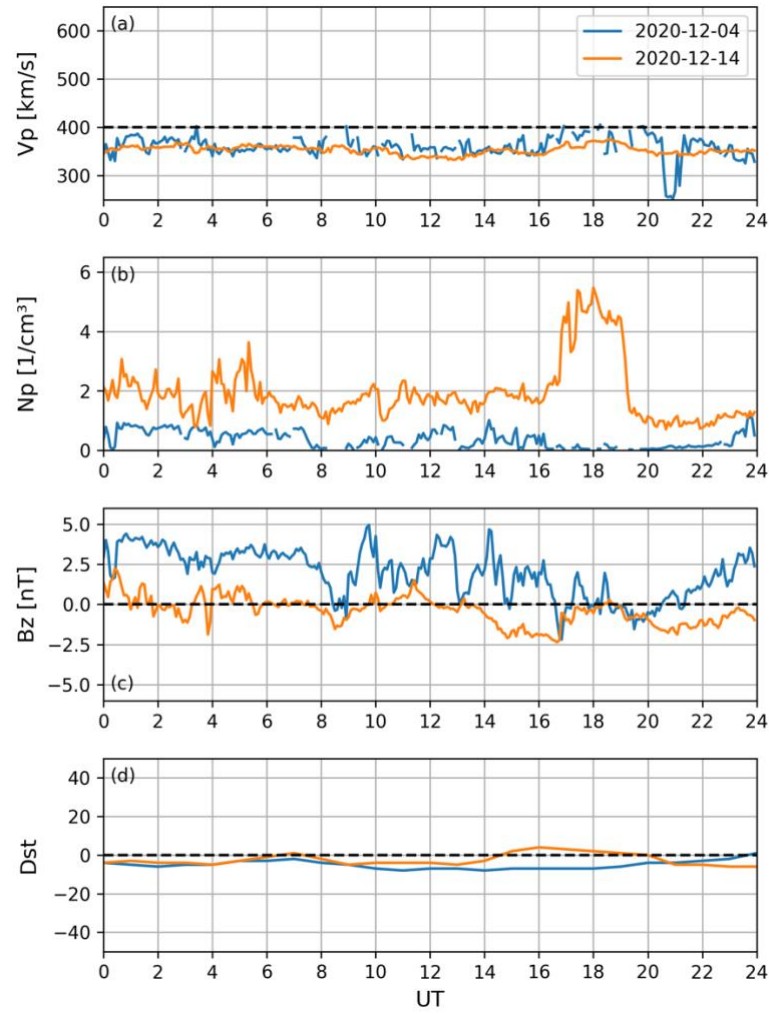

**Figure 2:** (a) The solar wind velocity Vp, (b) the number density of protons Np, (c) the interplanetary magnetic field component $B_z$, and (d) the Dst index on December 14, 2020 (orange line) and on December 04, 2020 (blue line).

### 3.2 Responses of the F region heights during the hours of the Solar Eclipse Event

5        In Figure 3, we investigate the minimum F layer virtual height $h'F$ (top) and F layer peak height $hmF2$ (bottom) to both analyzed regions in (a) CG and (b) CXP between 1400 UT and 1900 UT. The blue lines are the height parameters for the quietest day of the month (December 04, 2020), whereas the red lines refer to the solar eclipse day on December 14, 2020. The grey line means the solar eclipse obscuration for each region. At CG, the maximum obscuration occurred 1645 UT, reaching 20%. On the





other hand, the maximum obscuration occurred at 1715UT in CXP, with the most substantial value of 29%. Unfortunately, we do not have data over Santa Maria station (29.7º S, 53.8º W, dip ~ 37º S) during the solar eclipse hour occurrences, where the maximum obscuration reached 53%.

In both regions, we observe a strong fluctuation with high values of the $h`F$ before the solar eclipse onset. This behavior happens because a C4.0 class solar flare occurred between 14:09:00 and 14:56:00 UT, causing a radio blackout of the E and Es layer regions and partially the F region (Nogueira et al., 2015). The $h`F$ has low values concerning the quiet reference value after 1615 UT and 1700 UT for CG and CXP, respectively. At the same hours, a constant decrease was observed in the $hmF2$ parameter for these regions.

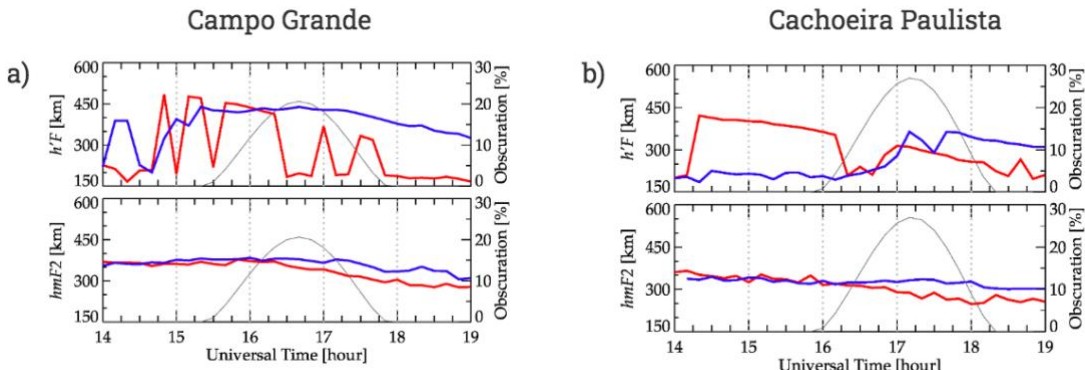

**Figure 3:** Virtual height $h`F$ (top) and F layer peak height $hmF2$ (bottom) in (a) CG and (b) CXP between 1400 UT and 1900 UT. These parameters are presented in blue line for the quietest day of the month (December 4, 2020), and in red line refers to the solar eclipse on December 14, 2020. The grey line means the solar eclipse obscuration for each region in percent.

To better observe this scenario, we show the ionograms for both regions. Figure 4 refers to CG on December 14, 2020, for 1500UT, 1730 UT, and 1750 UT. Notice that in hours before the solar eclipse, the F1 region is present, as the red arrow indicates. At 1730 UT, we observe that the F1 region has completely disappeared, returning at 1750 UT. As the peak obscuration for this station occurred around 1620 UT, the F1 region can suffer from the lost ionization, as discussed in Fargues et al. (2001). The same behavior seems to occur in CXP, as is shown in Figure 5. In this case, we observe the F1 layer at 1650 UT. Around 1710 UT, this layer disappears completely. However, a strong Es layer of "c" type (Es$_c$) caused by tidal winds (Resende et al., 2017a) appears also and can block the F region together. This





absence of the F1 region lasted until 1810 UT when this layer occurred again (as seen in the ionogram at
1820 UT). All these characteristics make the height profile of these regions decrease significantly, as we
saw in the *h`*F and *hm*F2 parameters in Figure 3.

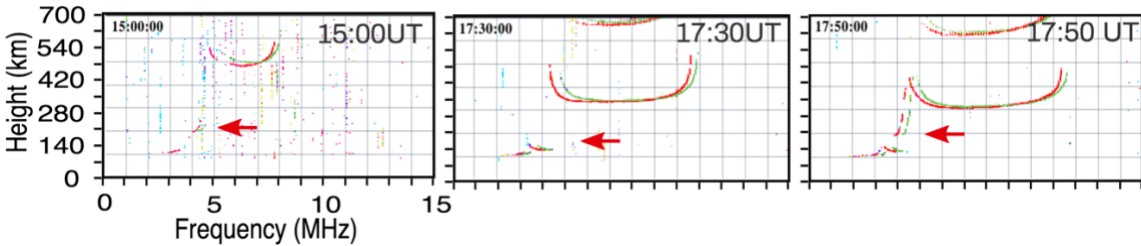

**Figure 4:** Ionograms collected at CG at 1500 UT, 1730 UT, and 1750 UT, showing the F1 layer disappearance during
the eclipse hours on December 14, 2020.

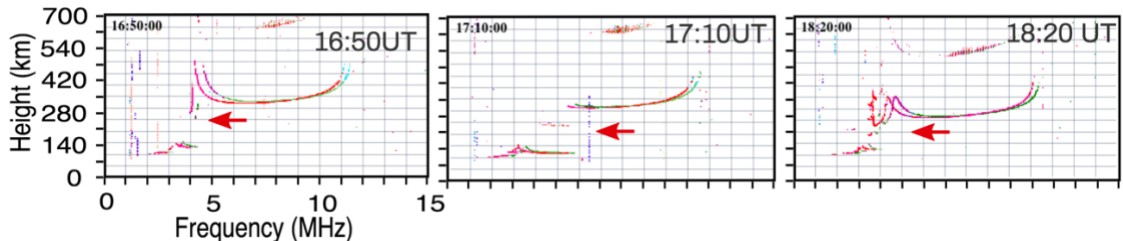

**Figure 5:** Ionograms collected at CXP at 1650 UT, 1710 UT, and 1820 UT, showing the F1 layer disappearance
during the eclipse hours on December 14, 2020.

Chandra et al. (2007) studied the ionospheric effects of the total solar eclipse of August 11, 1999,
over the Ahmedabad region (23° N, 73° E). The authors did not find any decrease in the critical frequency
of the F1 layer. However, Minnis (1955) analyzed the E and F1 layers during the solar eclipse of February
25, 1952. The author rewrote Chapman's equation, considering the fraction of the ionizing radiation lost
during the unobscured times. The theoretical results showed significant weakness in the F1 layer due to
the loss of electrons caused by an effective recombination process. More recently, Adeniyi et al. (2007)
showed the solar eclipse effect on the ionosphere over an equatorial station (8.53° N, 4.57° E, dip 4.1° S)
in the African region. This event was on March 29, 2006, and the maximum obscuration was 99 percent
in this station. One of their results was the evident absence of the F1 regions in ionograms over the station
analyzed. An explanation was that the electron density in the layers became so thin during the solar eclipse



events that the ionosonde could not detect it. However, unlike our result, the E region also disappears in Adeniyi et al. (2007). In Figures 4 and 5, the E and Es layers are evident, leading to uncertainties about the solar eclipse effect in the ionosphere on December 14, 2020. We believe here that the electron density decreases due to the recombination factor during the solar eclipse. The $Es_b$ layer simultaneous appearance

resulted in a significant weakening of this layer, making detection by the Digisonde difficult.

We do not observe significant differences in the $F_2$ layer densities in the ionograms data from these stations. Figure 6 shows the *fo*F2 parameter for CG (a) and CXP (b) during the reference day (blue line, December 04, 2020) and during the solar eclipse event (red line, December 14, 2020). The grey line means the solar eclipse obscuration for each region. Notice that over CG, the F region electron density

(related to the *fo*F2) has been smaller than the reference day since the previous hours of the solar eclipse event. Over CXP, the *fo*F2 values are practically the same on the two days analyzed. We credit this behavior to the low solar eclipse obscuration (20%-30%) over the ionospheric stations. The loss processes were insufficient to weaken the electron density at the F region, as seen in other events (Adeniyi et al., 2007).

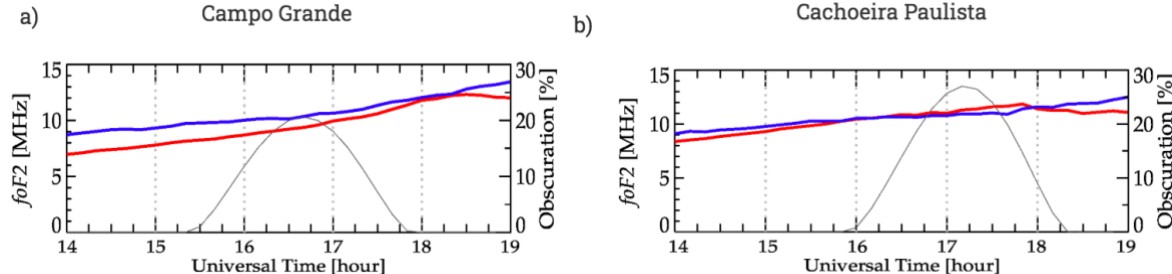

**Figure 6:** F region frequency (*fo*F2) in (a) CG and (b) CXP between 1400 UT and 1900 UT. These parameters are presented in blue line for the quietest day of the month (December 04, 2020), and in red line to the solar eclipse on December 14, 2020. The grey line means the solar eclipse obscuration for each region in percent.

The short ionization interruption could affect the EIA over the South American sector, as shown in Figure 7. This figure shows the TEC behavior through the maps (Takahashi et al., 2016), in which we have two high-density areas between 20°-30°S and 40°-60°W that characterizes the EIA. The red line refers to the magnetic equator, the circles (Figure 7c) refer to the eclipse area at 1700 UT, and the color scale in Figures 7a and 7b indicate the TEC intensity from 0 to 50 TECU (1 TECU = $10^{16}$ electrons/m²).

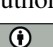



The EIA results from the equatorial plasma downward flows along magnetic field lines because of the diffusion and gravity. Therefore, two plasma crests are seen over the off-equatorial region, in the Northern and Southern Hemispheres (Nogueira et al., 2011). This behavior is evident in Figure 7a, when it is possible to observe the EIA peak around ±15° in TEC maps at 1700 UT, being stronger in the South regions. Notice that, on December 14, 2020 (Figure 7b), we observed a clear weakening of the EIA crests compared with the typical behavior of the ionosphere plasma (Figure 7a).

To better observe this difference, Figure 7c shows the relative difference (RD) parameter over the TEC maps computed by

$$RD\ (\%) = \left( (TEC_{Ref} - TEC_{SE})/TEC_{Ref} \right) \times 100. \tag{3}$$

The RD is calculated through the TEC maps for the solar eclipse day ($TEC_{SE}$) concerning the typical day ($TEC_{Ref}$). This result shows that the TEC is between 30% and 50% smaller during the eclipse occurrence over the Brazilian sector.

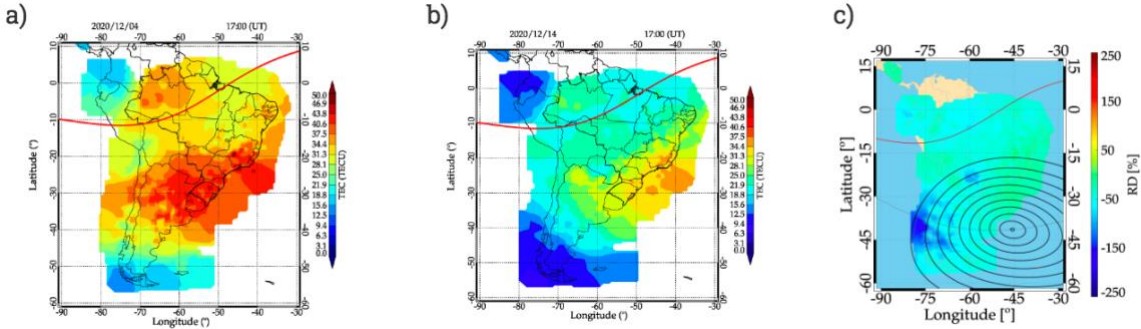

**Figure 7:** Longitude versus latitude distribution of the TEC map over South America at (a) during the reference period (December 04, 2020, at 1700 UT), (b) during the solar eclipse event (December 14, 2020, at 1700 UT) and (c) the RD parameter. The red line refers to the magnetic equator, the circle lines refer to the eclipse area, and the color scale indicates the TEC intensity.

Vyas and Sunda (2012) analyzed the TEC changes during an annular solar eclipse over the Indian sector on January 15, 2010. They showed a TEC reduction in the EIA localization that was named as inhibited EIA region. They attributed this behavior to the combined effects of the solar eclipse, which induce attenuation of EUV solar irradiation and the inhibited equatorial electrodynamics, affecting the



EIA. The negative deviation was 20-40% in the inhibited EIA region. Chen et al. (2019) modeled the EIA dynamic variations during a solar eclipse that occurred on August 21, 2017, around North and South America. They also obtained the TEC difference, and their results showed a significant reduction around EIA regions at solar eclipse times. Huang et al. (2020) studied the ionospheric responses at low latitudes in a solar eclipse on June 21, 2020. The authors also found that the EIA decreases significantly in the solar eclipse hours. In summary, Table 1 presents some recent studies that observed the EIA decrease during the solar eclipse events. Hence, we show the work reference, the solar eclipse date, and the percentage of the EIA decrease concerning the typical periods.

**Table 1:** Some studies that observed the EIA decrease during the solar eclipse events, as well as the percentage of the EIA layer decrease.

| Reference | Solar Eclipse Event | EIA decrease (%) |
|---|---|---|
| Vyas and Sunda (2012) | January 15, 2010 | ~ 20–40 |
| Chen et al. (2019a) | August 21, 2017 | ~ 40 |
| Jonah et al. (2020) | July 2, 2019 | ~ 35 |
| Huang et al. (2020) | June 21, 2020 | ~ 90 |
| Martínez-Ledesma et al. (2020) | December 14, 2020 (prediction) | ~22 |
| Our work | December 14, 2020 | 30-50 |

All the works cited before concluded that the ionization loss caused variation in the dynamic processes during the obscuration times, affecting the EIA behavior. The main hypothesis is that if the solar eclipse goes through the equatorial regions, the fountain effect will change, and consequently, less plasma density reaches the low latitudes. As predicted by Martínez-Ledesma et al. (2020), in our work, we believed that the minor TEC reduction around the magnetic equator locations was enough to cause a plasma density decrease in both EIA crests. Therefore, although this solar eclipse event almost did not reach the equatorial regions, we suppose it was enough to influence the fountain effect. Also, Huang et al. (2020) mentioned that during the eclipse events, the transequatorial northward/southward neutral wind



can weaken, causing a reduction of the EIA crests. In fact, to validate all these hypotheses, it is necessary to consider other equipment, which will be carried out in future work.

### 3.3 Responses of the E region and Es layers during the hours of the Solar Eclipse Event

The E region is dominated by the production and loss process in the ionosphere. Therefore, it is expected that the electron density of this layer suffers an influence during eclipses. Figure 8 shows the variation of the E region critical frequency parameter ($fo$E) in CG and (b) CXP between 1400 UT and 1900 UT. The blue lines refer to the quietest day of the month (December 4, 2020), and the red lines refer to the solar eclipse day on December 14, 2020. The grey line means the solar eclipse obscuration for each region. We note that the ionization starts to decrease around 1540 UT and 1550 UT for CG and CXP, respectively. Moreover, the values remain low until 1800 UT with respect to the quiet reference day.

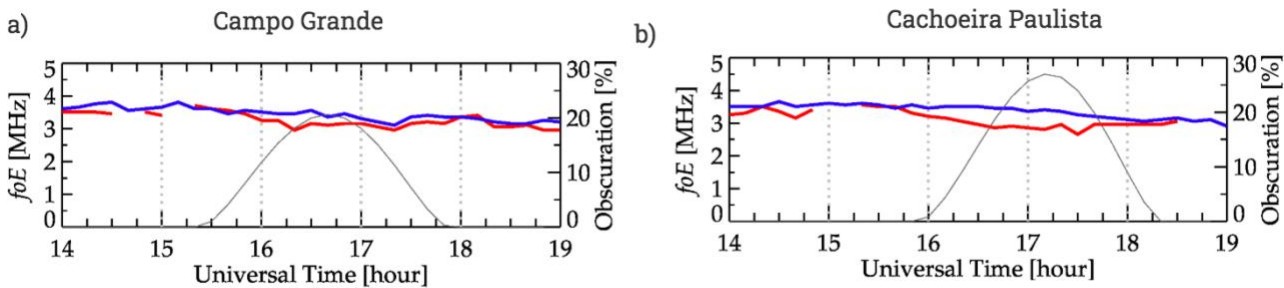

**Figure 8:** E region critical frequency ($fo$E) in (a) CG and (b) CXP between 1400 UT and 1900 UT. The parameters are presented in blue line for the quietest day of the month (December 04, 2020), and in red line refers to the solar eclipse on December 14, 2020. The grey line means the solar eclipse obscuration for each region in percent.

We observe a decrease of around 15% in the ionization in the E region during the solar eclipse obscuration (decrease from 4 MHz to ~ 3.3 MHz). Chernogor et al. (2019) analyzed the solar eclipse effects on March 20, 2015, on the mid-latitude daytime ionospheric plasma using observations from Kharkiv incoherent scatter radar (49.60° N, 36.30° E). They show that in heights less than 210 km, it is common to see a decrease in the electron density, mainly in the maximum phase of the solar eclipse. They found that the electron density reduced by 18.5 % in this event. Also, they believe that the explanation is related to the E region chemistry since the loss for recombination in these heights is quadratic. Thus, as





the ionization radiation from the Sun has been removed in this region, the loss processes become effective quickly, and the E region density is affected directly. Nonetheless, Rishbeth (1968) had already reported that during the partial or total eclipse events, the recombination process is not enough for the electron density to decay drastically. In fact, the authors mentioned that the duration time of the eclipse is not sufficient to affect the E region chemistry, making it disappear or diminish substantially. This fact can explain the reason for the E region electron density decrease 15% in our data.

Figure 9 shows the E region electron density simulated by MIRE in Height-Time-Intensity (HTI) maps over CG (left panel) and CXP (right panel). The profile background shows the regular E region, which is described by the significant electron density values in the daytime, and low values during the nighttime. The metallic ions and transport terms in Equations 1 and 2 were negligible since they are not important to analyze the E region. We have the E region profile considering the usual conditions in panel a, and, in panel b, we reduced the E region ionization by 15%.

As we expected, the results show an E region ionization reduction for both regions. We do not observe any differences in the E region behavior concerning the CG and CXP. In these two sites, the electron density maximum in the model decreased of 4.92 electrons/cm$^3$ for 4.87 electrons/cm$^3$ (in log scale) at around 1500 UT. Therefore, we show a significantly reduction in the E region electron density, which is mainly driven by chemical processes, that is directly affected by events such as eclipses, even if they are not total.



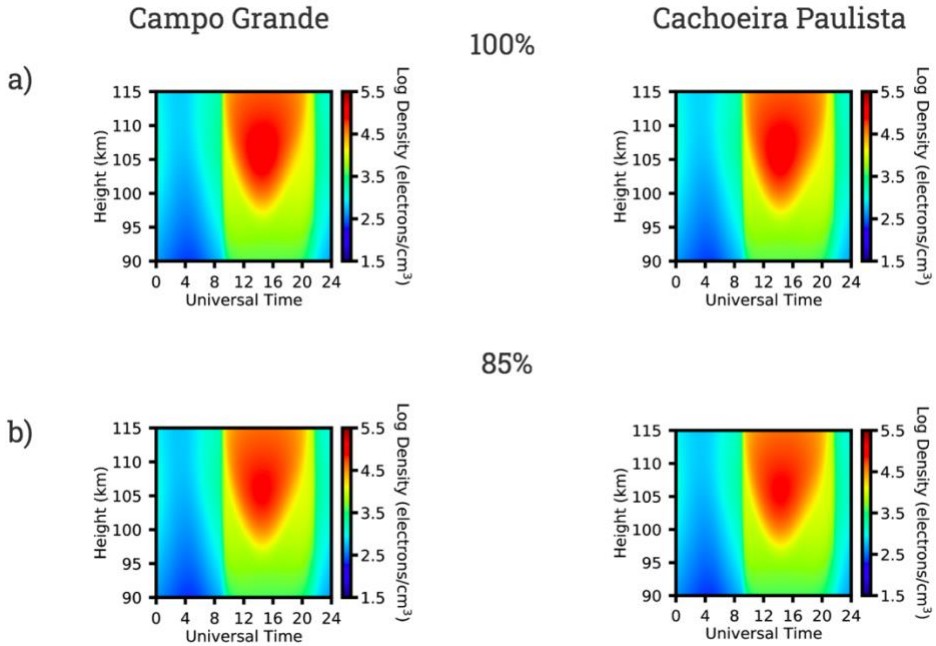

**Figure 9:** E region electron density simulated by MIRE considering the ionization for (a) 100% and (b) 85% on December 14, 2020, in Campo Grande (left panel) and Cachoeira Paulista (right panel).

Some works showed an intensification in the Es layers during eclipses. The main hypothesis is that atmospheric gravity waves can be induced during the total/partial solar eclipse and affect the vertical wind shear, strengthening the Es layer (Chen et al., 2010a, Chen et al., 2010b, Yadav et al., 2013). However, other works such as Pezzopane et al. (2015) showed that the solar eclipse did not affect the Es layer in terms of its intensity. In fact, the authors analyzed a partial effect of the solar eclipse that occurred on March 20, 2015, in mid-latitudes. Although their results did not show any Es layer intensity modification, they observed an evident influence in its time duration. The Es layer lasted longer, and they attributed this effect to the traveling ionospheric disturbances (TIDs) likely caused by gravity wave propagation.

Therefore, we evaluated the *fb*Es parameter over (a) CG and (b) CXP, as shown in Figure 10. The blue line refers to the quiet period reference and the red line to the solar eclipse day. In CG, the Es layer did not occur at almost any time of the day, which can be related to the weak wind that could not form the Es layers in this period (Resende et al., 2017a). Thus, it is not possible to analyze the Es layer in this




region. On the other hand, over CXP, we noted an interesting behavior: a peak in the *fb*Es at around 1710 UT and 1730 UT. To better analyze this fact, we show the ionogram (c) at 1720 UT, indicating that this parameter reached values higher than 6 MHz. After these hours, it returns to the typical values around 4 MHz (not shown here).

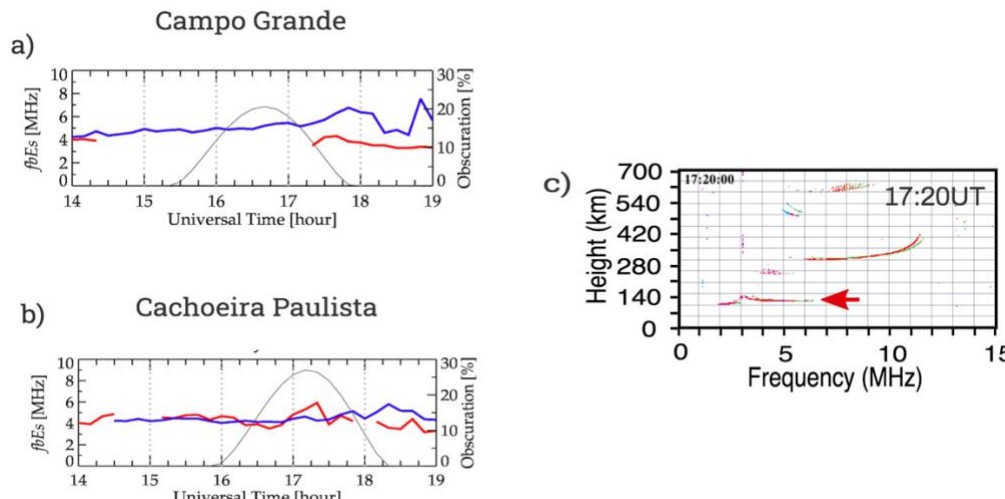

**Figure 10:** The blanketing frequency of the Es layer (*fb*Es) in (a) Campo Grande and (b) Cachoeira Paulista between 1400 UT and 1900 UT, and (c) the ionogram for Cachoeira Paulista at 1720 UT. These parameters are presented in blue line for the quietest day of the month (December 04, 2020), and in red line refers to the solar eclipse on December 14, 2020. The grey line means the solar eclipse obscuration for each region.

The Es layer seen over CXP is type "c", very common for this region and related to the zonal component of the wind. As we observe a clear increase of this layer electronic density, the hypothesis that the gravity waves influenced the winds becomes plausible. To verify if the gravity waves occurred, we use the Digisonde data as described in Abdu et al. (2009). The upper panel of Figure 11 shows the temporal variations in the true height of the ionogram fixed frequencies of 5, 6, and 7 MHz over CXP. Notice that an oscillation between 1300 UT and 1900 UT is apparent, indicating the presence of gravity waves. In the bottom panel of the same figure, we plotted the d($h$F) using a band-pass filter (30 min - 3 h) to remove considerable F region height gradients. We can observe a well-noticed in-phase oscillation during the eclipse hours (~1600 and 1800 UT). This fact means that there was gravity wave propagation in the E region that reached the F region. Also, in the dashed lines during the eclipse hours, we observe



clear gravity waves propagating upward. Thus, the gravity waves may have intensified the Es layer over CXP, as proposed by Chen et al.(2010a) and Chen et al.(2010b).

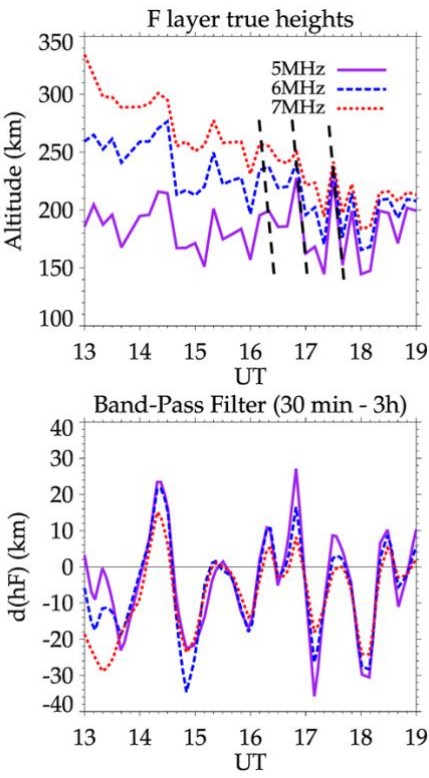

**Figure 11:** Temporal variations in the fixed frequency (5, 6, and 7 MHz) true heights, $h$F (upper panel), and deviation
of $h$F (band-pass filtered (30 min - 3 h)) (lower panel) in CXP on December 14, 2020.

## 4 Conclusions

This work analyzed the effects of the solar eclipse on December 14, 2020, over the Brazilian sector. At CG, the maximum obscuration occurred at 1645 UT, reaching 20%, and in CXP the maximum obscuration occurred at 1715UT, with the most substantial value of 29%. Also, this event happened during a quiet period, and consequently, it was possible to observe the specific feature of the solar eclipse effects in the ionosphere. The results showed that solar eclipses can cause significant ionosphere modifications even though they only partially reach the Brazilian low latitude regions. Finally, the main conclusions are summarized below:





1. The $h`F$ has low values concerning the quiet reference value after 1615 UT and 1700 UT for CG and CXP, respectively. At the same hours, a constant decrease was observed in the $hm$F2 parameter for these regions.

2. The F1 layer can suffer from the lost ionization during the solar eclipse day and disappears completely in some hours in both regions. However, the E and Es layers continued occurring. Thus, we believe that the electron density decrease is caused by the recombination factor, and the appearance of the $Es_b$ layer together resulted in a significant weakening of the E region, making detection by the Digisonde difficult.

3. We did not observe differences in the F2 layer densities in the ionograms data for these stations. We suppose that since these regions had a low solar eclipse obscuration (20%-30%), the loss processes were not sufficient to destabilize the F region, as seen in other events. However, the short ionization interruption can affect the EIA over the South American sector. The RD result showed that the TEC is between 30% and 50% smaller during the eclipse occurrence over the Brazilian sector. We believe that the minor TEC reduction around the magnetic equator locations was enough to cause a plasma density decrease in both EIA crests. Thus, although this solar eclipse event almost did not reach the equatorial regions, we assume it was enough to influence the fountain effect. We will further investigate this behavior in future works.

4. We observe a decrease of around 15% in the ionization in the E region. Additionally, the modeled results show an E region ionization reduction for both locations. We do not observe any differences in the E region behavior concerning the CG and CXP. In these two sites, the electron density maximum in model decreased of 4.92 electrons/cm$^3$ for 4.87 electrons/cm$^3$ (in log scale) at around 1500 UT. Therefore, as is already known, the E region is mainly driven by chemical processes and is directly affected by events such as eclipses, even if they are not total.

5. We observe an interesting behavior: a peak in the $fb$Es at around 1710 and 1730 UT in CXP. As proposed by previous works, we concluded that there was a gravity wave propagation in the E region, intensifying the Es layer over CXP.




**Acknowledgments**

L. C. A. Resende would like to thank the China-Brazil Joint Laboratory for Space Weather (CBJLSW),
National Space Science Center (NSSC), Chinese Academy of Sciences (CAS) for supporting her
postdoctoral. C. M. Denardini thanks CNPq/MCTI, grant 03121/2014-9. S. S. Chen thanks CNPq/MCTI
(grant 303643/2017-0). L. A. Da Silva, J. P. Marchezi, P. Jauer and D. Silva would like to thank the
CBJLSW/NSSC/CAS for supporting their postdoctoral. C. S. Carmo thanks CNPq/MCTI, Brazil (grant
141935/2020-0). J. Moro would like to thank the CBJLSW/NSSC/CAS for supporting his postdoctoral,
and the CNPq/MCTI (grant 429517/2018-01). D. Barros thanks CNPq/MCTI, grant 301988/2021-8. G.
A. S. Picanço thanks CNPq/MCTI, Brazil (grant 132252/2017-1) and Capes/MEC, Brazil (Grant
88887.351778/2019-00). R. P. Silva thanks CNPq/MCTIC, Brazil (grant 302000/2021-6).

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
