# Peer review of "A multi-instrumental and modelling analysis of the ionospheric responses to the solar eclipse of December 14, 2020, over the Brazilian region"

_Annales Geophysicae, 2021_

## Author Comment (AC1)

**Responses to the Comment and/or Suggestions from Referee 1**

Manuscript 'A multi-instrumental and modelling analysis of the ionospheric responses to the solar eclipse of December 14, 2020, over the Brazilian region' by Resende et al. submitted to the Annales Geophysicae.

General Comments

This paper describes the different aspects of ionospheric respose to solar solar eclipse in a very precise way. I strongly recommend this manuscript to get published in this journal. I really appreciate that authors have included the modelling study for E-region response to eclipse.

*Firstly, we appreciate the time the referee spent reviewing this article, and we thank the comments given by the referee. We have revised the manuscript taking into account all the referee's comments.*

Questions

1. Whether fmin values showing any changes?
*Our response:*
*In fact, during some hours of the solar eclipse (around 1700 UT), we observe that the F1 layer disappears completely in both analyzed regions (Campo Grande and Cachoeira Paulista). This behavior lasts around one hour, and we believe it is due to the solar eclipse event and the Es layer presence, which can block the F region together. We clarify this fact in the new version of the manuscript.*

2. Is there any wave activity observed in TEC data?
*Our response:*
*To answer the reviewer's question, we calculate the perturbation components of TEC (dTEC) from relative TEC (rTEC), subtracting the TEC trend obtained from a 1 h running average for a ground receiver and GNSS satellite as described in Figueiredo et al. (2017). The results were very interesting, showing the intense wave activity during the eclipse hours. In the figure below, we have presented the dTEC over Cachoeira Paulista (CPHI), and the wave activity is apparent. In fact, we noticed an increase in the amplitude of these waves in these hours. We will add this new analysis in the article as supplementary material.*

[Figure]

Reference: Figueiredo, C. A. O. B., C. M. Wrasse, H. Takahashi, Y. Otsuka, K. Shiokawa, and D. Barros (2017), Large-scale traveling ionospheric disturbances observed by GPS dTEC maps over North and South America on Saint Patrick's Day storm in 2015, J. Geophys. Res. Space Physics, 122, 4755–4763, doi:10.1002/ 2016JA023417.

Finally, we would like to take this opportunity to thank the reviewer for kindly evaluating our paper helping to greatly improve its quality.

---

## Author Comment (AC2)

**Responses to the Comment and/or Suggestions from Referee 2**

Manuscript 'A multi-instrumental and modelling analysis of the ionospheric responses to the solar eclipse of December 14, 2020, over the Brazilian region' by Resende et al. submitted to the Annales Geophysicae.

This paper studies the ionosphere response to the solar eclipse on Dec 14, 2020 using observations from ionosonde and TEC measurements, as well as MIRE numerical simulations. The F region and E region behavior are investigated accordingly. The authors attempt to highlight some interesting phenomena such as EIA reduction and Es enhancement, which could be an incremental contribution to the widely-carried subject of solar eclipse analysis. However, there are some issues with the current interpretation of observations that needs to be carefully addressed. Thus, a moderation revision is suggested before the paper can be recommended for publication. The detailed comments are as below.

Thank you very much for this important feedback. We appreciate the time the referee spent reviewing this article, and we thank the comments given by the referee. We answered your comments in the following.

1. It is nice to see that the authors add modeling results in their study since extensive solar eclipse studies are based merely on observations. There are some prior modeling studies of the solar eclipse that the authors can refer to or compare with.

Le, H., Liu, L., Yue, X., Wan, W., 2008. The ionospheric responses to the 11 August 1999 solar eclipse: observations and modeling. Ann. Geophys. 26, 107-116

Huba, J. D., and Drob, D. (2017). SAMI3 prediction of the impact of the 21 August 2017 total solar eclipse on the ionosphere/plasmasphere system. Geophysical Research Letters, 44, 5928-5935

Our response:

We thank the referee for suggesting these works. We added some important discussion using these references in our modeled results and some information in the Introduction.

2. Line 18-21, page 13. The major concern is the interpretation of the EIA crests reduction from TEC observations in Figure 7. The inhibition of the southern crest is of expectation since it is directly covered by eclipse obscuration with reduced photo-ionization. However, the TEC around the northern crest with no eclipse coverage also exhibited a large reduction. The authors claimed that "minor TEC reduction around the magnetic equator locations was enough to cause a plasma density decrease in both EIA crests". This description is a little bit misleading since the ionosphere has large day-to-day variability and it is arbitrary to simply ascribe the conjugate EIA reduction to the remote solar eclipse that did not cover the equatorial region. The authors are suggested to add TEC results for control days around eclipse (e.g., Dec 13 and Dec 15). This will provide important information to interpret the behavior of EIA crests.

**Our response:**

We understand the reviewer's concern, and the reviewer is right. We decided to choose the quietest day because, on days around the solar eclipse, we have the interplanetary magnetic field (IFM) sector boundary-crossing that could have caused some influence in the Es layer behavior over the Brazilian regions. However, as the referee suggested, we have already examined the days around the solar eclipse, and we found the same behavior for the south crest of the EIA. For the northern EIA crest on December 13, we also observed a decrease of the TEC, and on December 15, we did not observe any significant modification (please, see the figures below).

It is important to mention here that there are few GNSS receivers in the northern EIA crest. Thus, this fact needs further investigation in the future using other equipment, as we mentioned in this article. However, we include additional discussion in the article with the references cited in the next question.

3. The authors are suggested to be cautious in making such an arbitrary conclusion with no clear interpretation: "although this solar eclipse event almost did not reach the equatorial regions, we suppose it was enough to influence the fountain effect". What mechanism and how enough a remote solar eclipse can influence the equatorial fountain effect? The authors are suggested to make a preliminary discussion on possible reasons for EIA reduction for both crests, if they believe it is related to the solar eclipse. In fact, there are some prior studies (e.g., Le et al., 2009; Zhang et al., 2021; Aa et al., 2021; Huba et al., 2017) that discussed the possible mechanism in causing the conjugate TEC reduction, such as thermal cooling along the flux-tube, inter-hemispheric mapping of electric field, modified plasma pressure gradient and wind effect. The authors are suggested to refer to these studies to provide discussion. Of course they can validate the hypothesis in the future with new available equipment, but a preliminary interpretation and discussion are necessary.

Le, H., Liu, L., Yue, X., et al. 2009. The ionospheric behavior in conjugate hemispheres during the 3 October 2005 solar eclipse. Ann. Geophys. 27, 179-184

Zhang, S.-R., Erickson, P.J., Vierinen, et al., 2021. Conjugate ionospheric perturbation during the 2017 solar eclipse. J. Geophys. Res. Space Phys. 126.

Aa, E., Zhang, S.-R., Shen, H. et al., (2021). Local and conjugate ionospheric total electron content variation during the 21 June 2020 solar eclipse. Advances in Space Research, 68(8), 3435–3454

**Our response:**

We appreciate these references. In fact, we do not have an exact explanation for the EIA behavior. As the reviewer mentioned, an electron density reduction in the southern EIA crest reduction is expected since the solar eclipse reaches the area. However, we also observed weakness in the northern EIA crest. As mentioned in the text, Martínez-Ledesma et al. (2020) predict that the minor TEC reduction around the magnetic equator locations was enough to cause a plasma density decrease in both EIA crests. A plasma reduction over equatorial regions leads to less movement of this plasma to low latitudes in both hemispheres. Another hypothesis here is that the partial absence of the radiation over the Peruvian equatorial area in South America. Thus, there was a reduction of the equatorial density, and, consequently, the fountain effect can be affected by the solar eclipse. Also, another hypothesis is about the transequatorial northward/southward winds that change their configurations, affecting the ionosphere globally during the eclipse time.

All the articles cited by the reviewer are very interesting. We included them in the new version of this manuscript. Among them, Le et al. (2009) showed an analysis of the ionosphere in the conjugate hemisphere during the solar eclipse on October 3, 2005. Their main result is a decrease in the electron temperature in both conjugate points, which is associated with a reduction in the photoelectrons traveling along the magnetic field lines from the eclipse region to the conjugate region. Thus, the authors proposed that solar eclipse events can cause a disturbance in the ionospheric regions in the conjugate hemisphere. In such analysis, the TEC decrease around 32% in the 300 km. Recently, Zhang et al. (2021) analyzed the TEC perturbations in the southern/northern EIA crests of the solar eclipse on August 21, 2017. They found that in the southern crest of the anomaly, the TEC reduced significantly while in the northern crest stayed almost undisturbed. They mentioned that there is a northward motion tendency for plasma within the flux tubes that can inhibit the typical diffusion of the equatorial fountain effect. Therefore, as the reviewer suggested, we discuss this part with more details in this manuscript, adding the possible mechanism in causing the conjugate TEC reduction. We mentioned all these references and clarified our hypothesis in the manuscript.

4. Figure 9. It is hard to visually notify the difference between a and b. The authors are suggested to add differential results to make a better illustration.

**Our response:**

We included a contour value in this figure showing the electron density maximum in the model (~4.9 electrons/cm3 in 100% and 4.8 electrons/cm3 in 85%) (in log scale) (shown in the file/below).

---

## Author Comment (AC3)

**Responses to the Comment and/or Suggestions from Referee 2**

Manuscript 'A multi-instrumental and modelling analysis of the ionospheric responses to the solar eclipse of December 14, 2020, over the Brazilian region' by Resende et al. submitted to the Annales Geophysicae.

This paper studies the ionosphere response to the solar eclipse on Dec 14, 2020 using observations from ionosonde and TEC measurements, as well as MIRE numerical simulations. The F region and E region behavior are investigated accordingly. The authors attempt to highlight some interesting phenomena such as EIA reduction and Es enhancement, which could be an incremental contribution to the widely-carried subject of solar eclipse analysis. However, there are some issues with the current interpretation of observations that needs to be carefully addressed. Thus, a moderation revision is suggested before the paper can be recommended for publication. The detailed comments are as below.

*Thank you very much for this important feedback. We appreciate the time the referee spent reviewing this article, and we thank the comments given by the referee. We answered your comments in the following.*

1. It is nice to see that the authors add modeling results in their study since extensive solar eclipse studies are based merely on observations. There are some prior modeling studies of the solar eclipse that the authors can refer to or compare with.

Le, H., Liu, L., Yue, X., Wan, W., 2008. The ionospheric responses to the 11 August 1999 solar eclipse: observations and modeling. Ann. Geophys. 26, 107-116

Huba, J. D., and Drob, D. (2017). SAMI3 prediction of the impact of the 21 August 2017 total solar eclipse on the ionosphere/plasmasphere system. Geophysical Research Letters, 44, 5928-5935
*Our response:*
*We thank the referee for suggesting these works. We added some important discussion using these references in our modeled results and some information in the Introduction.*

2. Line 18-21, page 13. The major concern is the interpretation of the EIA crests reduction from TEC observations in Figure 7. The inhibition of the southern crest is of expectation since it is directly covered by eclipse obscuration with reduced photo-ionization. However, the TEC around the northern crest with no eclipse coverage also exhibited a large reduction. The authors claimed that "minor TEC reduction around the magnetic equator locations was enough to cause a plasma density decrease in both EIA crests". This description is a little bit misleading since the ionosphere has large day-to-day variability and it is arbitrary to simply ascribe the conjugate EIA reduction to the remote solar eclipse that did not cover the equatorial region. The authors are suggested to add TEC results for control days around eclipse (e.g., Dec 13 and Dec 15). This will provide important information to interpret the behavior of EIA crests.

*Our response:*

*We understand the reviewer's concern, and the reviewer is right. We decided to choose the quietest day because, on days around the solar eclipse, we have the interplanetary magnetic field (IFM) sector boundary-crossing that could have caused some influence in the Es layer behavior over the Brazilian regions. However, as the referee suggested, we have already examined the days around the solar eclipse, and we found the same behavior for the south crest of the EIA. For the northern EIA crest on December 13, we also observed a decrease of the TEC, and on December 15, we did not observe any significant modification (please, see the figures below).*

[Figure]

*It is important to mention here that there are few GNSS receivers in the northern EIA crest. Thus, this fact needs further investigation in the future using other equipment, as we mentioned in this article. However, we include additional discussion in the article with the references cited in the next question.*

3. The authors are suggested to be cautious in making such an arbitrary conclusion with no clear interpretation: "although this solar eclipse event almost did not reach the equatorial regions, we suppose it was enough to influence the fountain effect". What mechanism and how enough a remote solar eclipse can influence the equatorial fountain effect? The authors are suggested to make a preliminary discussion on possible reasons for EIA reduction for both crests, if they believe it is related to the solar eclipse. In fact, there are some prior studies (e.g., Le et al., 2009; Zhang et al., 2021; Aa et al., 2021; Huba et al., 2017) that discussed the possible mechanism in causing the conjugate TEC reduction, such as thermal cooling along the flux-tube, inter-hemispheric mapping of electric field, modified plasma pressure gradient and wind effect. The authors are suggested to refer to these studies to provide discussion. Of course they can validate the hypothesis in the future with new available equipment, but a preliminary interpretation and discussion are necessary.

Le, H., Liu, L., Yue, X., et al. 2009. The ionospheric behavior in conjugate hemispheres during the 3 October 2005 solar eclipse. Ann. Geophys. 27, 179-184

Zhang, S.-R., Erickson, P.J., Vierinen, et al., 2021. Conjugate ionospheric perturbation during the 2017 solar eclipse. J. Geophys. Res. Space Phys. 126.

Aa, E., Zhang, S.-R., Shen, H. et al., (2021). Local and conjugate ionospheric total electron content variation during the 21 June 2020 solar eclipse. Advances in Space Research, 68(8), 3435–3454

*Our response:*

*We appreciate these references. In fact, we do not have an exact explanation for the EIA behavior. As the reviewer mentioned, an electron density reduction in the southern EIA crest reduction is expected since the solar eclipse reaches the area. However, we also observed weakness in the northern EIA crest. As mentioned in the text, Martínez-Ledesma et al. (2020) predict that the minor TEC reduction around the magnetic equator locations was enough to cause a plasma density decrease in both EIA crests. A plasma reduction over equatorial regions leads to less movement of this plasma to low latitudes in both hemispheres. Another hypothesis here is that the partial absence of the radiation over the Peruvian equatorial sector changes the local conductivity and, probably, it affects the entire equatorial area in South America. Thus, there was a reduction of the equatorial density, and, consequently, the fountain effect can be affected by the solar eclipse. Also, another hypothesis is about the transequatorial northward/southward winds that change their configurations, affecting the ionosphere globally during the eclipse time.*

*All the articles cited by the reviewer are very interesting. We included them in the new version of this manuscript. Among them, Le et al. (2009) showed an analysis of the ionosphere in the conjugate hemisphere during the solar eclipse on October 3, 2005. Their main result is a decrease in the electron temperature in both conjugate points, which is associated with a reduction in the photoelectrons traveling along the magnetic field lines from the eclipse region to the conjugate region. Thus, the authors proposed that solar eclipse events can cause a disturbance in the ionospheric regions in the conjugate hemisphere. In such analysis, the TEC decrease around 32% in the 300 km. Recently, Zhang et al. (2021) analyzed the TEC perturbations in the southern/northern EIA crests of the solar eclipse on August 21, 2017. They found that in the southern crest of the anomaly, the TEC reduced significantly while in the northern crest stayed almost undisturbed. They mentioned that there is a northward motion tendency for plasma within the flux tubes that can inhibit the typical diffusion of the equatorial fountain effect. Therefore, as the reviewer suggested, we discuss this part with more details in this manuscript, adding the possible mechanism in causing the conjugate TEC reduction. We mentioned all these references and clarified our hypothesis in the manuscript.*

4. Figure 9. It is hard to visually notify the difference between a and b. The authors are suggested to add differential results to make a better illustration.

*Our response:*

*We included a contour value in this figure showing the electron density maximum in the model (~4.9 electrons/cm$^3$in 100% and 4.8 electrons/cm$^3$in 85%) (in log scale) (shown in the file/below).*

[Figure]

Figure 11. (1) How the slant dashed lines are derived? Do the authors mean that there are downward phase propagation trends to indicate the existence of gravity waves as indicated by Abdu et al. ? However, the peaks and valleys of three curves at different frequencies almost occurred at the same time. (2) Whether the oscillation is related to solar eclipse since it seems to be appeared before the local eclipse.

*Our response:*

*(1) The F layer true heights (hF) are obtained through the Digisonde data at specific plasma frequencies (5, 6, and 7 MHz). This methodology is the same used by Abdu et al. (2009) used to observe the gravity waves manifestation. The slant dashed lines refer to these waves' downward phase propagation (notice the peak for each frequency). Although a little inclination, these peaks have a difference for each frequency. And the band-pass filtered oscillations include periods from 30-min to 3 hours, revealing the oscillations, presenting downward phase propagation in these times.*

*(2) The referee is right when he/she mentioned that we have oscillations before the solar eclipse. However, during the solar eclipse hours, it seems that there was an intensification of the drift velocity. This fact explains the atypical Es layer in Cachoeira Paulista in these hours. In this work, we only show the gravity waves presence to be a possible cause of the Es layer density enhancement.*

> *To prove this statement mentioned above, we calculate the perturbation components of TEC (dTEC) from relative TEC (rTEC), subtracting the TEC trend obtained from a 1 h running average for a ground-based GNSS receiver as described in Figueiredo et al. (2017). The results showed the intense wave activity during the eclipse hours. The next figure shows the dTEC over Cachoeira Paulista (CHPI), and the wave activity is apparent. In fact, we noticed an increase in the amplitude of these waves in these hours. We will add this new analysis in the article as supplementary material.*

[Figure]

Reference: Figueiredo, C. A. O. B., C. M. Wrasse, H. Takahashi, Y. Otsuka, K. Shiokawa, and D. Barros (2017), Large-scale traveling ionospheric disturbances observed by GPS dTEC maps over North and South America on Saint Patrick's Day storm in 2015, J. Geophys. Res. Space Physics, 122, 4755–4763, doi:10.1002/ 2016JA023417.

Minor:

5. Line 27, page 2: electronic (electron)
   *Our response:*
   *Ok!*

6. Equation 3, page 12: Exchange TEC(SE) and TEC(ref) terms.
   *Our response:*
   *Done.*

7. Figure 7c. The color scale can be adjusted. It is hard to identify the positive/negative values between -50~50 with yellow-greenish color.
   *Our response:*
   *Ok! We replaced this figure with a more appropriate scale. We prefer to use -170 to 170 to observe the TEC reduction in South America.*

Finally, we would like to take this opportunity to thank the reviewer for kindly evaluating our paper.